# Octopamine signaling regulates the intracellular pattern of the presynaptic active zone scaffold within *Drosophila* mushroom body neurons

Hongyang Wu[1,ᵒ], Sayaka Eno[1,ᵒ], Kyoko Jinnai[2], Ayako Abe[1], Kokoro Saito[1], Yoh Maekawa[3], Darren W. Williams[4], Nobuhiro Yamagata[1,5], Shu Kondo[2], Hiromu Tanimoto[1,3*]

1 Graduate School of Life Sciences, Tohoku University, Sendai, Japan, 2 Department of Biological Science and Technology, Tokyo University of Science, Tokyo, Japan, 3 Department of Biology, Faculty of Science, Tohoku University, Sendai, Japan, 4 Centre of Developmental Neurobiology, Institute of Psychiatry, Psychology & Neuroscience, King's College London, London, United Kingdom, 5 Department of Life Science, Graduate School of Engineering Science, Akita University, Akita, Japan

ᵒ These authors contributed equally to this work.
* hiromut@m.tohoku.ac.jp

## Abstract

Neurons can adjust synaptic output according to the postsynaptic partners. However, the target-specific regulation of synaptic structures within individual neurons in the central nervous system remains unresolved. Applying the CRISPR/Cas9-mediated split-GFP tagging, we visualized the endogenous active zone scaffold protein, Bruchpilot (Brp), in specific cells. This technology enabled the spatial characterization of the presynaptic scaffolds only within the Kenyon cells (KCs) of the *Drosophila* mushroom bodies. We found the patterned accumulation of Brp among the compartments of axon terminals, where a KC synapses onto different postsynaptic neurons. Mechanistically, the localized octopaminergic projections along γ KC terminals regulate this compartmental Brp heterogeneity via Octβ2R and cAMP signaling. We further found that physiological stress, such as food or sleep deprivation reorganizes this intracellular pattern in an octopamine-dependent manner. Such concurrent regulation of local active zone assemblies thus suggests how the mushroom bodies integrate changing physiological states.

## Introduction

The molecular composition of chemical synapses underlies the function of neurons and therefore exhibits a remarkable diversity among cell types [1,2]. As demonstrated by studies on the motor neuron, presynaptic molecular assemblies can be highly heterogeneous even within a single cell [3,4]. Critically, the difference of presynaptic structures may result in varying synaptic functions at the single active zone (AZ) level, such as release probability [5–7]. Given that a single neuron in the central

**Data availability statement:** All relevant data are within the paper and its Supporting Information files (Excel file named "S1 Data").

**Funding:** This work was supported by the following grants: Japan Society for the Promotion of Science 25H00986 (HT) Funder website: https://www.jsps.go.jp/english/ Japan Society for the Promotion of Science 22H05481 (HT) Funder website: https://www.jsps.go.jp/english/ Japan Society for the Promotion of Science 22KK0106 (HT) Funder website: https://www.jsps.go.jp/english/ Japan Society for the Promotion of Science 20H00519 (HT) Funder website: https://www.jsps.go.jp/english/ Japan Science and Technology Agency JPMJSP2114 (HW)\ Funder website: https://www.jst.go.jp/EN/ Biotechnology and Biological Sciences Research Council (BBSRC) BB/T013869/1 (DW) Funder website: https://www.ukri.org/councils/bbsrc/ Tohoku University Research Program 'Frontier Research in Duo' (HT) Funder website: https://w3.tohoku.ac.jp/frid-en/ The funders had no role in study design, data collection and analysis, decision to publish, or preparation of the manuscript.

**Competing interests:** The authors have declared that no competing interests exist.

**Abbreviations:** APF, after puparium formation; AZ, active zone; Brp, Bruchpilot; Cac, Cacophony; cAMP, cyclic adenosine monophosphate; CNS, central nervous system; KC, Kenyon cell; MBs, mushroom bodies; MBONs, MB output neurons; OANs, octopaminergic neurons; PSF, point spread function; RNAi, RNA interference; Rut, Rutabaga; STED, stimulated emission depletion.

nervous system (CNS) typically synapses onto multiple target cells, the spatial adaptation of output machineries could differentiate the activities of postsynaptic cells. While such intracellular synaptic heterogeneity influences complex computation of the circuit in the CNS, the target-specific AZ regulation is scarcely studied [8].

To this end, neurons projecting to the *Drosophila* mushroom bodies (MBs) can serve as an excellent model system. Kenyon cells (KCs), the major MB intrinsic neurons, synapse onto five sets of post-synaptic partners in spatially segregated compartments [9,10]. Postsynaptic MB output neurons (MBONs) have distinct and compartmentalized dendrites within the MB lobes [10], and their activities collectively determine the MB output [11–15]. The compartmental distinctions in presynaptic assemblies within individual KCs are likely to be critical in controlling MB-guided behaviors. However, it is challenging to characterize AZs only in KCs without 3D electron microscopy, because the synapse density in MB lobes is among the highest in the fly brain [16,17].

We here aim to characterize the compartmental distinctions of presynaptic structures of KCs by visualizing endogenous Brp in a cell-type-specific manner. *Drosophila* ELKS/CAST/ERC family member Brp is a major scaffold protein forming the T-shape electron-dense projections decorating AZs [18–21]. Brp plays a central role in molecular assemblies at AZs by accumulating calcium channels and synaptic vesicles [18,22]. Therefore, Brp enrichment serves as a proxy for estimating the synaptic vesicle release probability at single AZs [5,7]. To label endogenous presynaptic proteins in designated cell types [23,24], we took advantage of the CRISPR/Cas9-mediated split-GFP tagging strategy to target Brp in the dense network of MBs [25,26]. Profiling the character of individual Brp clusters only within KCs revealed the compartmental Brp accumulation pattern along the MB lobes. We further report the state-dependent remodeling of this intracellularly organized AZ structure and the regulatory mechanisms.

## Results

### Cell-type-specific visualization of endogenous active zone scaffold proteins Brp

Chemical tagging using Brp::SNAP [27] suggests endogenous Brp localizes heterogeneously within the MB (S1 Fig), which is constituted by pre-synapses of various cell types (S2 Fig). To visualize endogenous Brp only in designated cell types, we inserted the $GFP_{11}$ fragment (the 11th β-strand of the super-folder GFP) just prior to the stop codon of *brp* using CRISPR/Cas9. The self-assembly and reconstitution of GFP is induced by expressing the $GFP_{1–10}$ fragments using the GAL4/UAS system (Fig 1A). Stimulated emission depletion (STED) super-resolution microscopy revealed donut-shape Brp::rGFP accumulation at individual AZs of motor neuron terminals in the larva, confirming the undisturbed localization and molecular assembly of Brp::rGFP (S3 Fig) [18].

We validated this system in the CNS with GAL4 driver lines that label 7 different MB-projecting neuronal populations: 3 KC subtypes, 2 dopamine neuron clusters

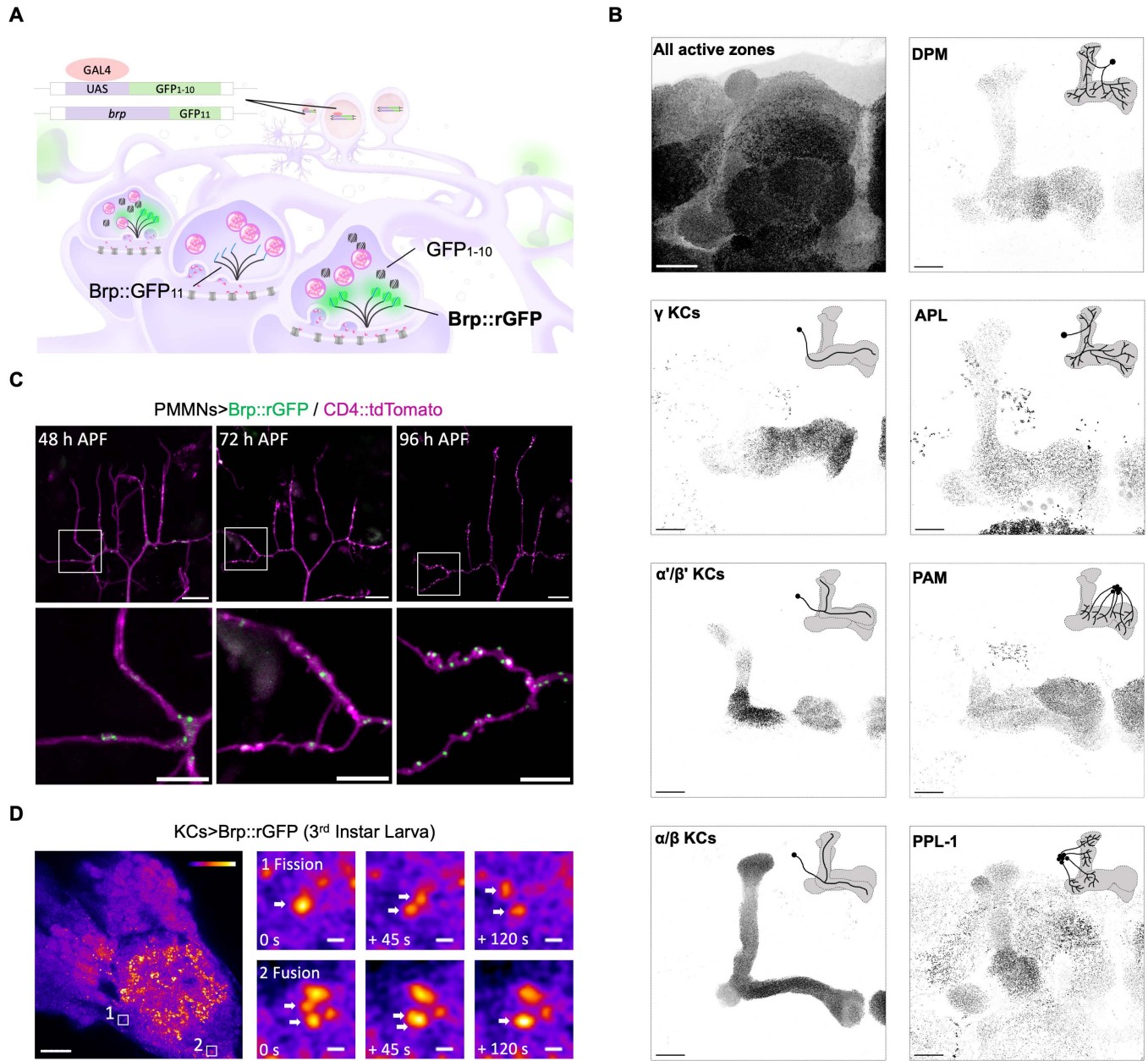

**Fig 1. Cell-type-specific dissection of the endogenous active zone scaffold. (A)** Schematic of split-GFP tagging system. GFP$_{11}$ was inserted prior to the stop codon of *brp*, while GFP$_{1-10}$ was expressed in specific cell types using the GAL4-UAS system. GFP$_{11}$ and GFP$_{1-10}$ reconstitute to emit fluorescence. **(B)** Visualization of Brp::rGFP in a cell-type-specific manner. Maximum intensity projections are shown. Immunostaining using antibody nc82 which labels all Brp in the brain, as a comparison to Brp::rGFP. Diagrams in each panel show the innervation pattern of neurons in the MB. GAL4 lines used: γ KCs (*MB009B-GAL4*), α/β KCs (*MB008B-GAL4*), α'/β' KCs (*MB370B-GAL4*), PPL-1 (*TH-GAL4*), PAM (*R58E02-GAL4*), DPM (*VT64246-GAL4*), APL (*GH146-GAL4*). Scale bar, 20 μm. **(C)** Long-term live imaging of Brp::rGFP (green) in a growing motor neuron (magenta, CD4::tdTomato) from 48 to 96 h APF. *OK371-GAL4* was used to express GFP$_{1-10}$ and CD4::tdTomato. White boxes indicate zoomed-in areas shown in the lower panels. Scale bars: upper panels, 10 μm; lower panels, 5 μm. **(D)** *Ex vivo* live-imaging of Brp::rGFP in 3rd instar larval KCs. GFP$_{1-10}$ was expressed using *R13F02-GAL4*. The left panel shows the MB calyx region. White boxes indicate zoomed-in areas shown on the right panels. Boxes 1 and 2 demonstrate Brp::rGFP fission and fusion event. White arrows indicate Brp::rGFP clusters undergoing fission or fusion. Scale bars: left panel, 10 μm; right panels, 200 nm.

(PPL1 and PAM) and 2 single interneurons (DPM and APL neurons) [10,28]. Confocal microscopy detected reconstituted GFP (rGFP) signals only at restricted areas, in contrast to the dense signals in Brp::SNAP or anti-Brp immunostaining (Figs 1B and S1). The Brp::rGFP signal was localized to axon terminals of given cell types and present in discrete particles (Fig 1B). Especially, Brp can be labeled at single-cell resolution for the APL and DPM neuron [9,29,30]. Strikingly, the Brp::rGFP signal intensity appeared to be heterogeneous along the axon terminals of KCs (Fig 1B). Since individual KCs arborize onto the entire lobe structure, these results suggest distinct accumulations of Brp intracellularly.

To test whether Brp::rGFP labels dynamic synaptic structures, we benchmarked its performance with different live-imaging contexts. The axon arborizations of pleural muscle motor neurons (PM-Mns) can be imaged on the abdominal body wall during pupal development [31]. Live-imaging Brp::rGFP in the PM-Mn revealed the formation of individual clusters at 48 hours after puparium formation and the increasing density in the same branch following two days (Fig 1C). We further visualized Brp::rGFP dynamics in the *ex vivo* preparation of the larval brain, and found multiple fusion and fission events of Brp clusters in the MB calyx (Fig 1D). These results are consistent with the AZ assembly through liquid–liquid phase separation [32] and validate the method to visualize the dynamics of endogenous Brp structures.

## Intracellular active zone heterogeneity among MB compartments

Brp::rGFP accumulation was heterogeneous among KC compartments, especially in γ KCs (Figs 2A, 2B and S4). As this difference in Brp::rGFP signal intensity cannot be explained by technical issues, such as imaging depth (S5 Fig), it is likely that Brp accumulation varies by compartments in individual KCs (Fig 2B). To further substantiate these intracellular differences in Brp accumulation, we visualized Brp::rGFP clusters within single KCs by stochastically expressing $GFP_{1–10}$ using flp-out GAL4 [33]. Measurement of GFP intensity of discrete clusters in single γKCs revealed significant differences of Brp clusters among compartments (Fig 2C and 2D). This result suggests that KCs tune their presynaptic AZ structures according to postsynaptic partners (Fig 2B).

In motor neurons, Brp is well characterized to serve as a scaffold protein accumulating other synapse proteins such as voltage-gated $Ca^{2+}$ channels [18,20,34]. Consistently, we found that the endogenous α subunit of voltage-gated $Ca^{2+}$ channel, Cacophony (Cac) co-localizes with Brp in the γ lobe (S6 and S7 Figs). Given the compartmental Brp heterogeneity (Fig 2A–2D), this correlation suggests that the $Ca^{2+}$ concentration is differentially set along the γ KCs. To measure the basal $Ca^{2+}$ concentration nearby AZs, we expressed $Brp^{short}$::mCherry::GCaMP6s, a ratiometric $Ca^{2+}$ sensor fused to mCherry and truncated Brp [35], in KCs. We live-imaged the MB and found that basal GCaMP signals were compartmentally distinct and gradually increased towards the γ5 compartment (Fig 2E). This pattern was reproducible using the Brp-independent $Ca^{2+}$ sensor myr::GCaMP6s [5] normalized to the reference signal CD4::tdTomato (S8 Fig), and similar to Brp::rGFP intensity pattern within individual γKCs (Fig 2C and 2D). Taken together the colocalization of Brp and Cac, these results suggest that the Brp accumulation per AZ is set compartmentally, differentiating the basal $Ca^{2+}$ influx via $Ca^{2+}$ channels at AZs [36].

## The state-dependent regulation of Brp compartmental heterogeneity

To efficiently quantify the Brp heterogeneity in γKCs, we established an image-processing pipeline for measuring signal intensity of individual Brp::rGFP clusters. Optimizing the parameter combination of 3D image deconvolution [37,38] and 3D spot segmentation enabled us to resolve Brp::rGFP clusters that represent single AZs using conventional confocal microscopy [30]. We measured the median GFP intensity of Brp::rGFP clusters in a volume sampled from each compartment of γKCs (Fig 3A). As this measurement was applied to individual puncta, the result of this analysis is independent of the AZ density or the compartmental differences of AZ numbers (S9 Fig). The analysis revealed a striking compartmental heterogeneity of AZs (Fig 3B). The median Brp::rGFP intensity of clusters in the γ5 compartment was nearly three times as intense as that in γ1. To quantify the heterogeneity levels of Brp clusters, we calculated the variance of the

**A**

KCs>Brp::rGFP

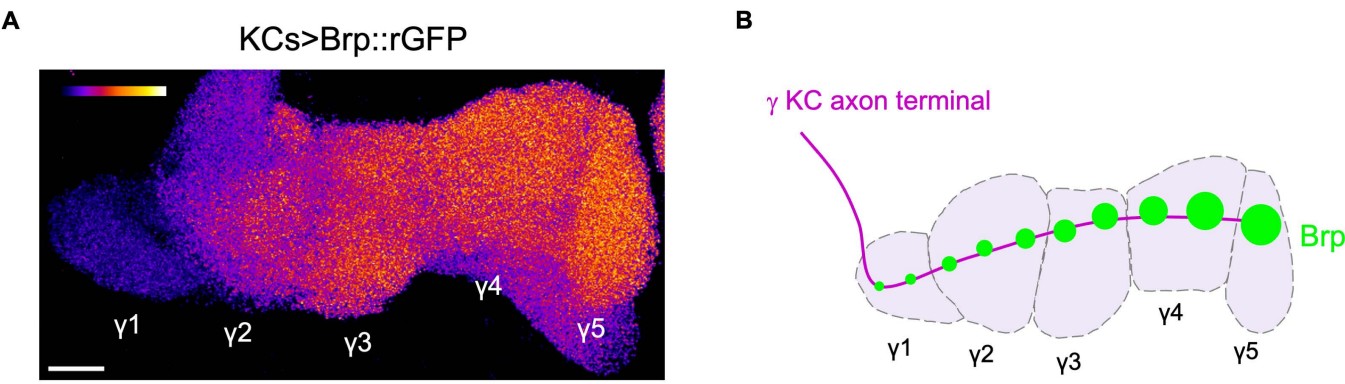

**B**

γ KC axon terminal

Brp

**C**

Single γ KC>Brp::rGFP / CD4::tdTomato

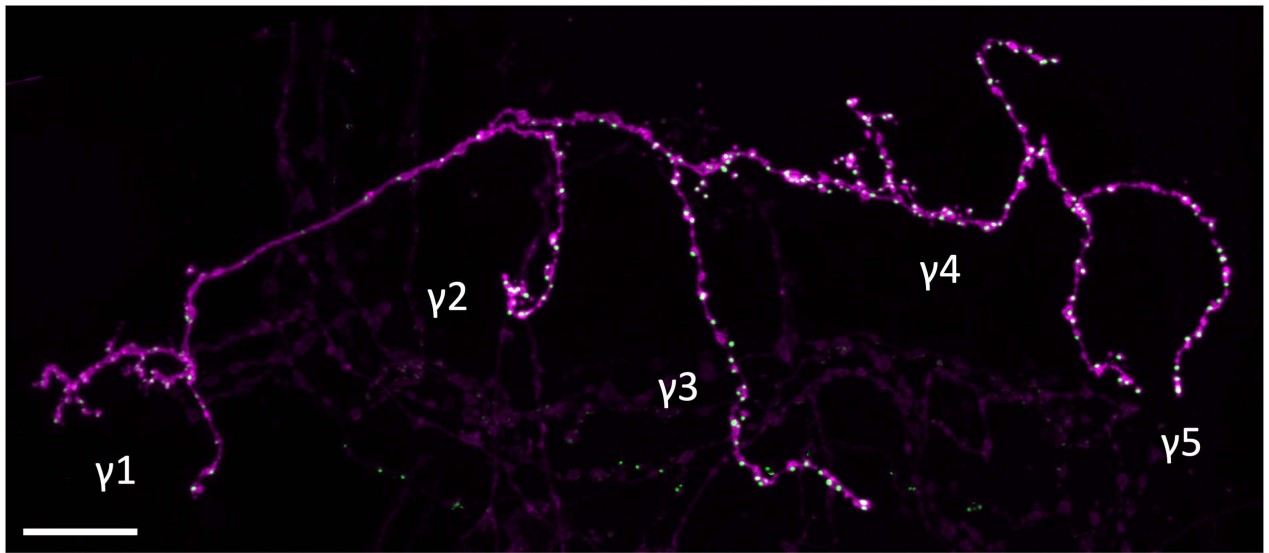

**D**

Brp::rGFP in single KCs

**E**

KCs>Brp^short^::mCherry::GCaMP6s live imaging

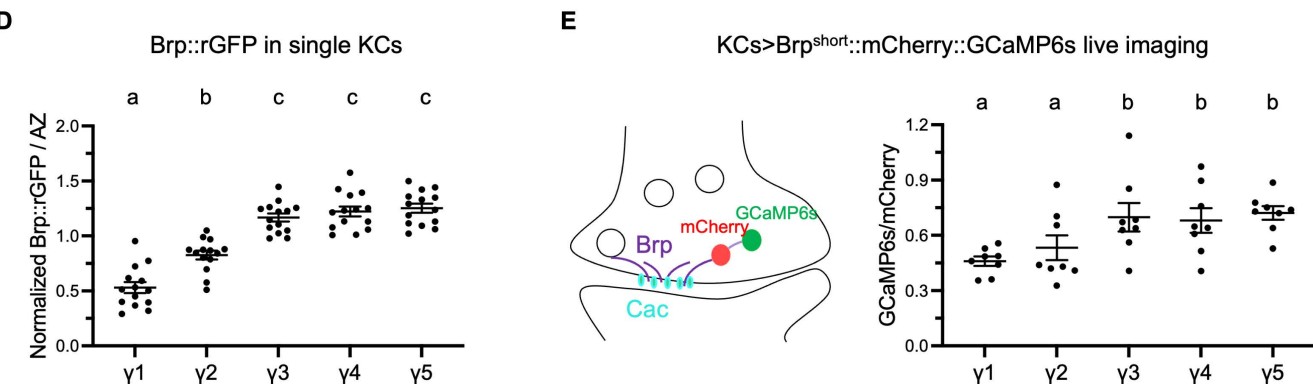

**Fig 2. Intracellular active zone heterogeneity among MB compartments. (A)** Brp::rGFP visualized in KCs. Maximum intensity projection showing the horizontal lobe of the MB. Brp::rGFP was visualized using *R13F02-GAL4*. **(B)** Diagram showing compartments in the MB γ lobe and the intracellular Brp localization in a single γ KC. **(C)** Representative maximum intensity projection visualizing Brp::rGFP and CD4::tdTomato in a single γ KC. **(D)** Signal intensity of Brp::rGFP clusters of single KCs in different compartments. Individual Brp::rGFP clusters are measured. The graph shows median Brp::rGFP

intensities of compartments, normalized to the average of the five compartments' medians. $n = 14$ KCs (one KC per mushroom body, 9 brains). Repeated measures one-way ANOVA. γ1 vs. γ2: $P = 0.0004$; γ1 vs. γ3: $P < 0.0001$; γ1 vs. γ4: $P < 0.0001$; γ1 vs. γ5: $P < 0.0001$; γ2 vs. γ3: $P < 0.0001$; γ2 vs. γ4: $P = 0.0001$; γ2 vs. γ5: $P < 0.0001$. **(E)** Basal $Ca^{2+}$ concentration near the active zones varies by compartments. The left panel shows the schematic of the Brp::mCherry::GCaMP6s ratiometric sensor. GCaMP6s sensor is fused to mCherry and Brp$^{short}$. The sensor was expressed by *R13F02-GAL4* and GCaMP6s signal is normalized by mCherry for analysis. Repeated measures one-way ANOVA. $n = 8$ brains. γ1 vs. γ3: $P = 0.0317$; γ1 vs. γ4: $P = 0.0307$; γ1 vs. γ5: $P < 0.0001$; γ2 vs. γ3: $P = 0.0150$; γ2 vs. γ4: $P = 0.0307$; γ2 vs. γ5: $P = 0.0150$; Scale bars, 10 μm. Error bars show S.E.M. Significant differences ($P < 0.05$) are indicated by distinct letters. The data underlying this Figure can be found in S1 Data.

log-transformed compartmental medians within each sample (Fig 3C), making the measure independent of between-sample differences in Brp::rGFP intensity.

As different MB compartments are functionally coordinated and integrate internal states such as nutritional states, sleep need and aging [11,30,39–43], we examined whether KCs adapt the synaptic structures upon physiological changes. To test the state-dependent structural plasticity of intracellular Brp accumulation, we compared the Brp::rGFP heterogeneity levels in γ KCs among fed, 48-hour starved and refed flies (Fig 3D). Strikingly, the compartmental heterogeneity in γ KCs significantly decreased upon food deprivation and recovered after refeeding (Figs 3D and S12A). These results suggest that feeding stress can drive local adjustment of AZ structures within KCs, reflecting reorganized compartmental activities.

## Octopamine input underlies the Brp compartmental heterogeneity

Each compartment receives dopamine projections from structurally and functionally distinct subsets of dopaminergic neurons (S10A Fig) [10]. We therefore examined whether dopamine inputs determine the compartmental Brp accumulation by knocking-down dopamine receptors (DopR1, DopR2 and D2R) in KCs using RNA interference (RNAi) [44,45]. None of these disruptions showed a significant effect on the compartmental heterogeneity (S10B and S12B Figs).

The octopaminergic neurons (OANs), particularly OA-VPM3 and OA-VPM4 neurons, are known to project to the MB and concentrate their innervations in the γ1 compartment [10,46]. We characterized presynaptic components of OA-VPM3/4 visualized by Brp::rGFP and nSyb::CLIP (neuronal synaptobrevin, nSyb; synaptic vesicle marker) and found they were densely localized to γ1 compared to the other compartments, complementary to the Brp compartmental heterogeneity of KCs (Fig 4A–4D). These results suggest localized synaptic output of OANs along the γ lobe, potentially underlying the Brp compartmental heterogeneity.

To examine the contribution of octopamine to the Brp accumulation within KCs, we measured the Brp compartmental heterogeneity in the mutants of tyramine β-hydroxylase (*Tβh*), the enzyme that catalyzes the synthesis of octopamine from tyramine [47]. We generated two new *Tβh* null alleles: insertion mutant *Tβh$^{SK1-8}$* and deletion mutant *Tβh$^{SK2-4}$* using CRISPR/Cas9-mediated mutagenesis. We visualized Brp::rGFP in KCs of these mutants and found the significantly decreased compartmental heterogeneity of the γ KCs (Figs 4E and S12C), likely due to the failure of pattern establishment during early adulthood (S11 Fig). This suggests that octopamine directly controls the AZ structure in KCs.

To identify the signaling pathway that regulates the Brp compartmental heterogeneity, we knocked-down octopamine/tyramine receptors (including OAMB, Octα2R, Octβ1R, Octβ2R, Octβ3R, OctTyrR, TyrR, TyrRII) in KCs using RNAi [44,48]. Among all the receptors tested, only Octβ2R knockdown significantly decreased the compartmental heterogeneity (Figs 5A, S12D and S12E). We could not test the effect of Octβ1R and Octβ3R because their deficiency in KCs caused abnormal development of the MB. Octβ2R is coupled to the $G_s$ alpha subunit to stimulate cyclic adenosine monophosphate (cAMP) production [49]. To test whether cAMP plays a role in Brp accumulation, we knocked-down the adenylate cyclase Rutabaga (Rut) in KCs using RNAi [44,50–54]. The downregulation of Rut resulted in a significant decrease of the compartmental heterogeneity (Figs 5B and S12F), suggesting that octopamine functions through Octβ2R–cAMP pathway.

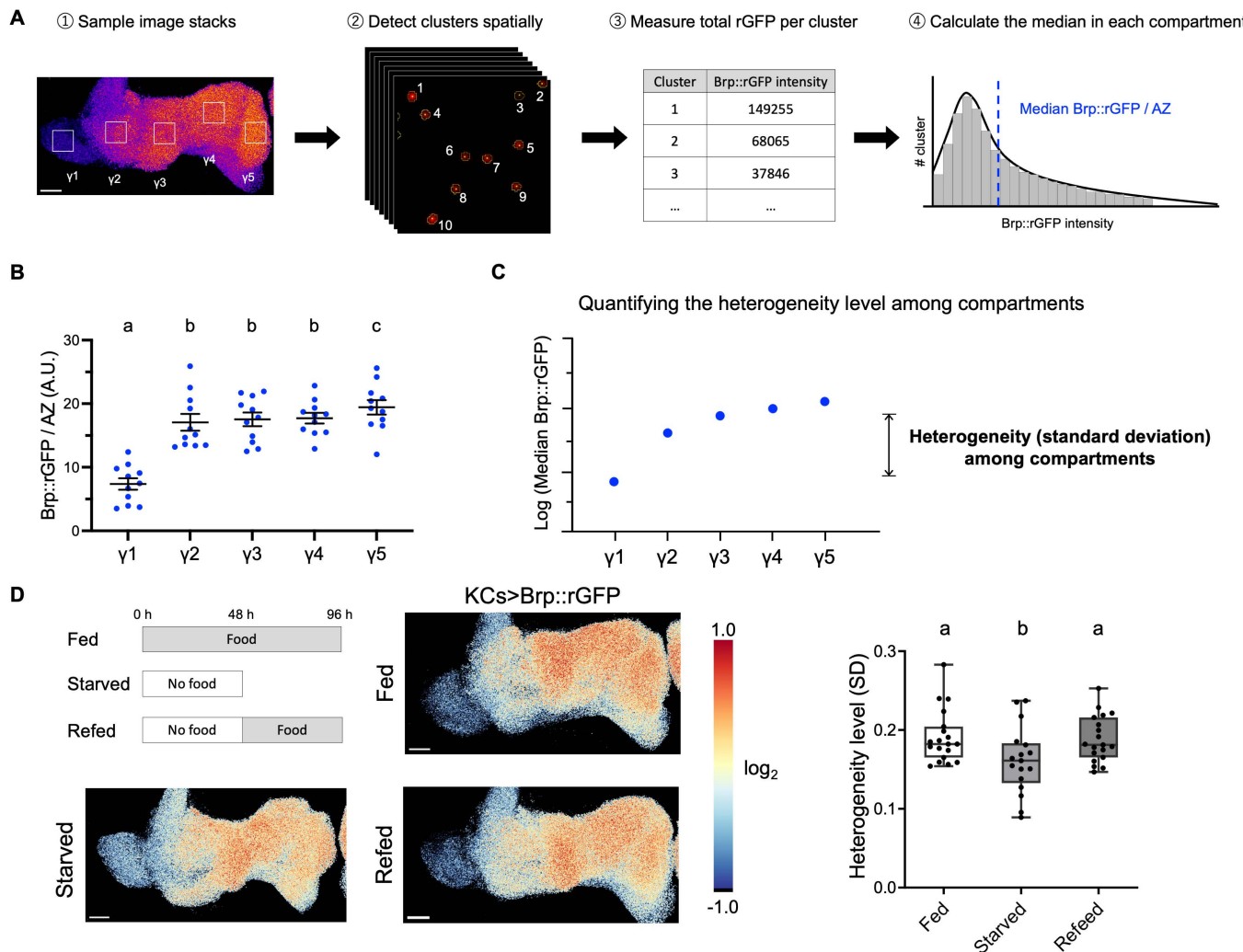

**Fig 3. Acute starvation decreases the Brp::rGFP heterogeneity level among γcompartments. (A)** Graphical summary of the image analysis. 3D volumes were sampled from different γ compartments. Brp::rGFP clusters are then spatially detected using the 3D spot segmentation Fiji plugin. rGFP signal of individual clusters was measured and the median intensity is calculated for each compartment. **(B)** Median intensity of Brp::rGFP clusters in γ lobe compartments. *R13F02-GAL4* was used. Repeated measures one-way ANOVA. n = 11 brains. γ1 vs. γ2: *P* < 0.0001; γ1 vs. γ3: *P* < 0.0001; γ1 vs. γ4: *P* < 0.0001; γ1 vs. γ5: *P* < 0.0001. γ2 vs. γ5: *P* = 0.0101; γ3 vs. γ5: *P* = 0.0162; γ4 vs. γ5: *P* = 0.0038. Error bars show S.E.M. **(C)** Standard deviation (SD) as an indicative of Brp::rGFP heterogeneity level among compartments. The median of Brp::rGFP intensity of clusters in each compartment was log-transformed and SD was calculated from five log-transformed medians for each brain sample. **(D)** Starvation for 48 hours reduced the Brp heterogeneity level in KCs. One-to-two-week-old flies were used for this experiment. Pseudo color in the images represents the value of $\log_2$ (pixel intensity/mean pixel intensity in the γ lobe). Pseudo color range: −1.0 to 1.0. The heterogeneity level (SD) was quantified for individual brain samples. One-way ANOVA on log-transformed data. Fed (*n* = 19 brains) vs. Starved (*n* = 17 brains): *P* = 0.0167; Starved vs. Refeed (*n* = 19 brains): *P* = 0.0167; Fed vs. Refeed: *P* = 0.8271; Scale bar, 10 μm. Box plots showing center (median), whiskers (Min. to Max.). Significant differences (*P* < 0.05) are indicated by distinct letters. The data underlying this Figure can be found in S1 Data.

## Internal states adjust the Brp compartmental heterogeneity through octopamine signaling

The spontaneous firing of octopamine neurons was shown to be decreased upon starvation [55]. Therefore, we hypothesized state-dependent AZ remodeling through octopamine (Fig 6A). To test this hypothesis, we starved the *Tβh* mutants and measured Brp::rGFP heterogeneity levels in γ KCs. Indeed, there was no significant state-dependent plasticity in the

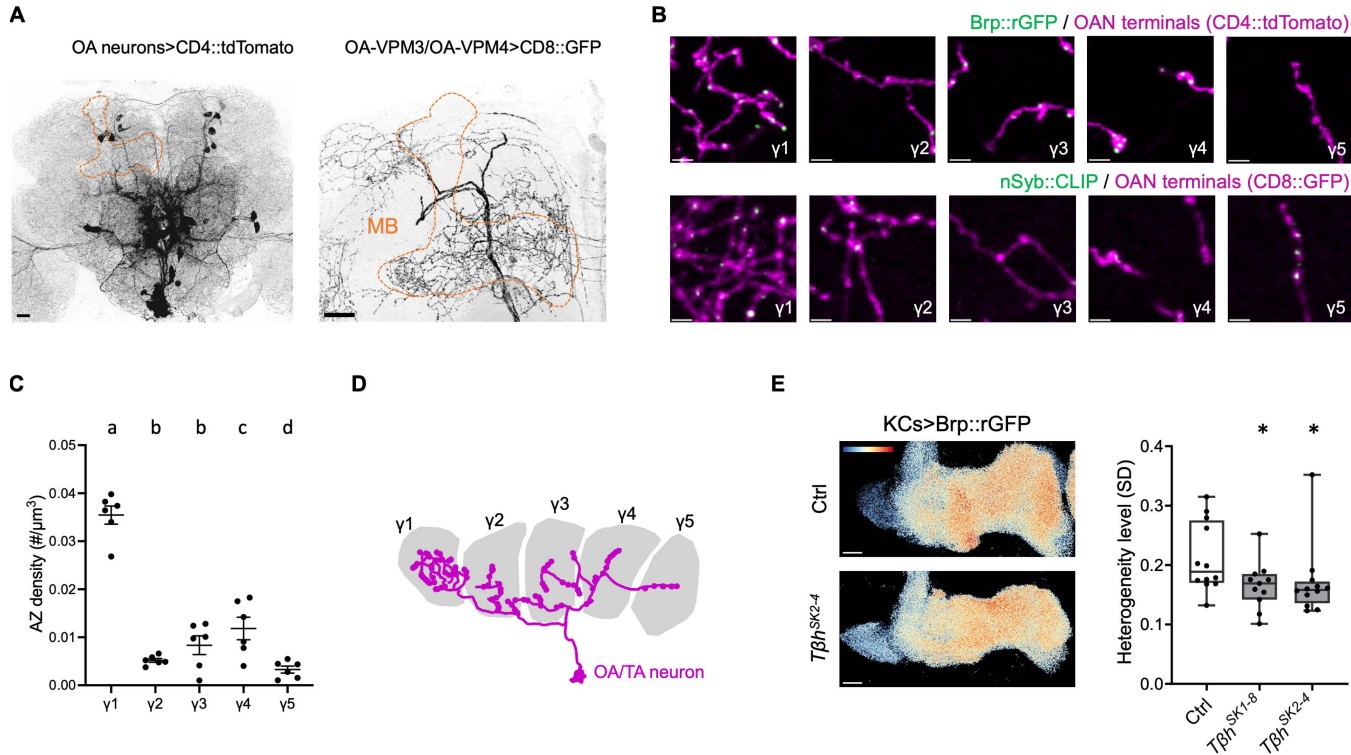

**Fig 4. Octopamine controls the AZ structure in KCs. (A)** OANs in the adult brain. OANs were labeled by *Tdc2-GAL4* driven CD4::tdTomato in the left panel. OA-VPM3/4 neurons were labeled with *MB022B-GAL4* [10] driven CD8::GFP (right panel). Dashed lines indicate the MB. Scale bars, 20 μm. **(B)** AZ number and synaptic vesicle localization of OANs along the γ lobe. $GFP_{1-10}$ and CD4::tdTomato were expressed using *Tdc2-GAL4*. nSyb::CLIP labeling synaptic vesicles and CD8::GFP were expressed using *MB022B-GAL4*. Scale bars, 2 μm. Images were cropped from the same MB and are identical in size. **(C)** AZ density of OANs across γ compartment. Brp::rGFP was visualized using *Tdc2-GAL4*, and cluster density was quantified. $n = 6$ brains. Repeated measures one-way ANOVA. γ1 vs. γ2: $P < 0.0001$; γ1 vs. γ3: $P < 0.0001$; γ1 vs. γ5: $P < 0.0001$; γ1 vs. γ5: $P < 0.0001$. γ2 vs. γ4: $P = 0.0098$; γ3 vs. γ5: $P = 0.0407$; γ4 vs. γ5: $P = 0.0015$. Error bars show S.E.M. **(D)** Schematic showing the innervation pattern of OANs along the γ lobe. **(E)** Reduced Brp compartmental heterogeneity level in *Tβh* mutants. Representative images of Brp::rGFP in control and *Tβh^{SK2-4}* background are shown. Pseudo color in the images represents the value of $\log_2$ (pixel intensity/mean pixel intensity in the γ lobe). Pseudo color range: −1.0 to 1.0. Scale bars, 10 μm. Kruskal–Wallis test. Ctrl ($n = 12$ brains) vs. *Tβh^{SK1-8}* ($n = 11$ brains) and *Tβh^{SK2-4}* ($n = 12$ brains). Ctrl vs. *Tβh^{SK1-8}*: $P = 0.0442$; Ctrl vs. *Tβh^{SK2-4}*: $P = 0.0228$. *$P < 0.05$. Box plots showing center (median), whiskers (Min. to Max.). Significant differences ($P < 0.05$) are indicated by distinct letters or *. The data underlying this Figure can be found in S1 Data.

*Tβh* mutants, while γ KCs in the control flies showed starvation-induced decrease in Brp heterogeneity levels (Figs 6B and S12 and S12H). Furthermore, we found that sleep loss, another form of stress associated with AZ structural plasticity [42,56], modulated the Brp::rGFP heterogeneity levels, and again requiring *Tβh* (Figs 6C and S12I and S12J). Taken together, we propose that octopamine signaling adjusts the synaptic structures within KCs in response to changing internal states.

## Discussion

By labeling endogenous AZ scaffold protein Brp, we resolved individual AZs at the single-cell level. The substantial intracellular heterogeneity of Brp accumulation within γ KCs is found to be regulated by internal states of the animal. Other than the compartmental pattern of Brp clusters within KCs, our experimental pipeline enables further profiling of other AZ characteristics such as the local AZ density [30].

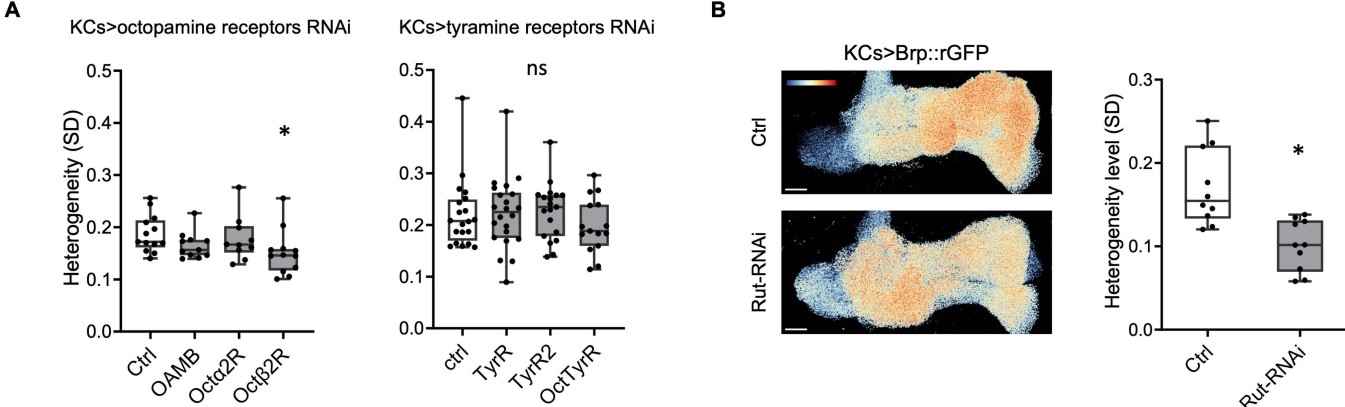

**Fig 5. Octβ2R and cAMP underlie the Brp compartmental heterogeneity. (A)** Knockdown of Octβ2R reduced the Brp heterogeneity level in γ KCs. Receptors are knocked-down in γ KCs specifically using RNAi with *R13F02-GAL4*. Kruskal–Wallis test. Ctrl ($n = 13$ brains) vs. OAMB ($n = 11$ brains), Octα2R ($n = 10$ brains) and Octβ2R ($n = 12$ brains). $P = 0.0228$ for Ctrl vs. Octβ2R. Ctrl ($n = 20$ brains) vs. TyrR ($n = 22$ brains), TyrR2 ($n = 22$ brains) and OctTyrR ($n = 15$ brains). *$P < 0.05$; ns = not significant. **(B)** Knockdown of Rut significantly decreased the Brp heterogeneity level in γ KCs. Representative images showing Brp::rGFP in control and in Rut knockdown group. Pseudo color in the images represents the value of log2(pixel intensity/mean pixel intensity in the γ lobe). Pseudo color range: −1.2 to 1.2. Mann–Whitney *U*-test. Ctrl ($n = 10$ brains) vs. Rut-RNAi ($n = 10$ brains): $P = 0.0011$. Box plots showing center (median), whiskers (Min. to Max.). *$P < 0.05$ and ns = not significant. The data underlying this Figure can be found in S1 Data.

How does octopaminergic signaling modify Brp accumulation? This study identified the requirement of Octβ2R and Rut in regulating the AZ structure in KCs (Fig 5). Octβ2R was shown to stimulate cAMP synthesis [49], suggesting that cAMP could play a determining role. Consistent with this idea, *rut* mutant displays abnormal AZ structures [57]. We propose that the localized innervations of octopamine neurons onto KCs (Fig 4) result in different cAMP concentrations along the γ lobe. In both KCs and motor neurons, cAMP concentration was shown to be locally regulated [58–60]. The affinity or stability of Brp to AZs among different compartments is perhaps distinctly set by the cAMP signaling compartmentation [61]. Interestingly, both DopR1 and Octβ2R stimulate cAMP biosynthesis, while the knock-down of DopR1 had no significant effect on the Brp heterogeneity (S10 Fig). This may be due to different downstream targets of the receptors including PKA [62].

AZ structures, especially Brp clusters, frequently serve as a proxy to explain synaptic functions [5–7,18,20,34,63]. Consistent with previous studies, our data suggest that Brp regulates the localization of Ca²⁺ channels in KCs [18,20,64]. The Brp heterogeneity may result in compartmental basal Ca²⁺ concentrations at AZs (Figs 2E and S8) and thus modulate spontaneous synaptic vesicle release via the stochastic activity of Ca²⁺ channels [36,65]. Indeed, the enrichment of the Ca²⁺ channel is associated with higher spontaneous release frequency in *Drosophila* neuromuscular junctions [7]. Although the relationship between Brp accumulation and evoked release in KCs seems more complicated since it involves acute and compartmental dopaminergic modulations [14,66–68]. Unlike evoked activities, spontaneous release of synaptic vesicles persists in the absence of stimulation thereby affecting the states of the animal [11].

The Brp compartmental heterogeneity within KCs might directly modulate the MB output through controlling the activity of different MBONs. Each type of MBON serves as an independent output unit, together regulating a variety of behaviors [12–15,42,69–71]. Starvation changes the activity balance between MBON-γ1 and MBON-γ5, thus enhances appetitive memory expression [69,71,72], and MBON-γ5 and MBON-γ2 oppositely control sleep amount [73]. Consistently, we found that the compartmental AZ heterogeneity within KCs, measured by Brp, decreased upon food or sleep deprivation (Fig 6B and 6C). Considering the contributions of octopamine to both feeding behaviors [55,74–80] and sleep control [81], KCs influence behavioral adaptations in response to changing internal states and environments by local AZ tuning [42,82,83].

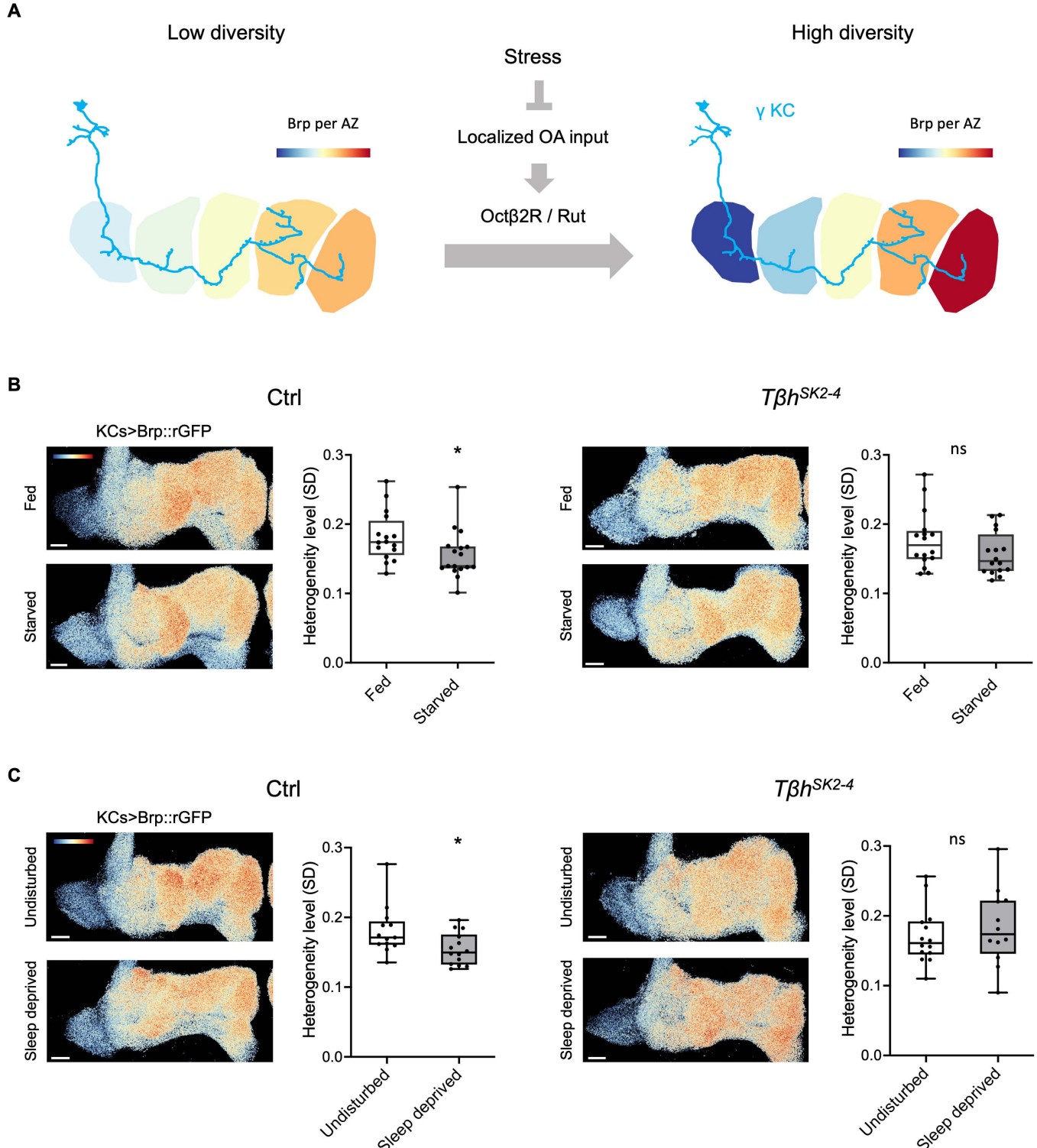

**Fig 6. Internal states adjust the Brp compartmental heterogeneity through octopamine. (A)** Graphical summary of the working model. Local-ized innervation of OANs onto γ KCs regulates the Brp accumulation and forms the Brp compartmental heterogeneity via Octβ2R and cAMP pathway. Nutritional states may modulate OAN activity, thereby adjusting the Brp heterogeneity. **(B)** Starvation for 48 hours did not affect the Brp::rGFP hetero-geneity level in *Tβh^SK2-4^* mutant. GFP$_{1-10}$ was expressed in KCs using *R13F02-GAL4* in control, or *Tβh^SK2-4^* background flies. 1−2 weeks old flies were

used. Pseudo color in the images represents the value of $\log_2$ (pixel intensity/mean pixel intensity in the γ lobe). Pseudo color range: −1.2 to 1.2. For the wild-type (WT) group, Fed ($n = 16$ brains) vs. Starved ($n = 17$ brains): $P = 0.0187$; For the $T\beta h^{SK2-4}$ group, Fed ($n = 16$ brains) vs. Starved ($n = 17$ brains): $P > 0.05$. Mann–Whitney $U$-test. **(C)** Sleep deprivation for 12 h decreases Brp::rGFP heterogeneity level in a $T\beta h$-dependent manner. For the Ctrl group, Undisturbed ($n = 13$ brains) vs. Sleep deprived ($n = 14$ brains): $P = 0.0222$; For the $T\beta h^{SK2-4}$ group, Undisturbed ($n = 14$ brains) vs. Sleep deprived ($n = 12$ brains): $P > 0.05$. Mann–Whitney $U$-test. Pseudo color range: −1.1 to 1.1. Box plots showing center (median), whiskers (Min. to Max.). $*P < 0.05$ and ns = not significant. The data underlying this Figure can be found in S1 Data.

## Materials and methods

### Animal husbandry

Flies were maintained on standard cornmeal food at 25 °C under a 12:12 h light-dark cycle for all experiments. Unless otherwise specified, 3–7 days old adult males were used for all experiments. For the starvation experiments, 1–2 weeks old flies were used for the starvation experiments. After hatching, adult flies were transferred to fresh food vials and flipped every 2–3 days before experiments. In experiments where flies were fasted or sleep deprived, the same population of flies was separated just before food deprivation for 48 h or sleep deprivation for 12 h. For sleep deprivation, flies in a vial were vibrated with a vortex mixer for 0.3 s at a random time point in every 5 s. Deprivation started from the light-off time in the light-dark cycle and continued for 12 h. In the flp-out GAL4 experiment for labeling single KCs, 3rd instar larvae were heat-shocked at 37 °C for 30 min in a water bath. $brp::GFP_{11}$ animals in this study were heterozygous. Strains used in this study are as indicated in Table 1.

### Transgenic lines and mutants

Two independent null alleles of the $T\beta h$ gene were obtained by inducing frameshift indels by the transgenic CRISPR/Cas9 system as previously described [94]. The following 20-bp gRNA sequences were used:

gRNA#1: GATACGTACACCAGTCCGGA

gRNA#2: GGTGAGACGGGACTACCAGC

$T\beta h^{SK1-8}$ was derived from mutagenesis by gRNA#1 and carries the following frameshift insertion: CGTACACCAGgatggacactGGATGGACAG (the inserted sequence is indicated by lower-case characters). $T\beta h^{SK2-4-4}$ was derived from mutagenesis by gRNA#2 and carries the following frameshift deletion: TCCGGATGGA----------------------------CTGTGAGGTC (the gap is indicated as hyphens). The $brp::GFP_{11}$ line was generated by targeted insertion of a $GFP_{11}$ cassette into the endogenous $brp$ locus just prior to the stop codon as previously described [26].

### Sample preparation

Animals from control and experimental groups were dissected on the same day. Data were collected from multiple batches of experiments performed on different days. All steps were performed at room temperature unless otherwise specified. Flies were anesthetized on ice and placed on ice before dissection. The dissection was performed in ice-cold PBS solution. Brain samples are fixed by 2% paraformaldehyde for 1 hour then washed 3 × 20 min with 0.1% PBT (0.1% Triton X-100 in PBS) in a PCR tube, typically containing 5–6 brains.

For SNAP or CLIP chemical tagging, brains were incubated in SNAP-Surface 647 (1:1000; NEB; S9136S) or CLIP-Surface 647 (1:1000; NEB; S9234S) in 0.3% PBT for 15 min and washed 3 × 20 min with 0.1% PBT. Samples are mounted using SeeDB2S, SeeDB2G [95] or 86% glycerol according to the imaging condition. Brp::SNAP, Brp::rGFP, Cac::sfGFP, CD4::tdTomato and CD8::GFP samples were imaged without immunohistochemistry.

Immunohistochemistry procedures (for anti-Brp staining in Fig 1B) were carried out as described [96]. In brief, fixed brains were washed 3 × 20 min with 0.1% PBT and incubated in 3% normal goat serum (NGS; Sigma-Aldrich;

**Table 1. Key resources used in this study.**

| Reagent or resource | Source | Identifier |
|---|---|---|
| Antibodies | | |
| Anti-Brp | DSHB | Cat#nc82; RRID: AB_2314866 |
| AlexaFluor-568 goat anti-mouse | Invitrogen | Cat#A11004; RRID: AB_2534072 |
| Chemicals | | |
| SNAP-Surface Alexa Fluor 647 | New England Biolabs | Cat#S9136S |
| CLIP-Surface Alexa Fluor 647 | New England Biolabs | Cat#S9234S |
| OMNIPAQUE 350 | GE Healthcare | Cat#22000AMX02447 |
| Histodenz | Sigma | Cat#D2158 |
| Tetraspeck Microspheres 0.1 mm | Thermo Fisher Scientific | Cat#T7279 |
| NaCl | Nacalai Tesque | Cat#31320-05 |
| KCl | Nacalai Tesque | Cat#28514-75 |
| TES | Nacalai Tesque | Cat#329810-55 |
| Trehalose dihydrate | Tokyo Kasei Kogyo | Cat#T0331 |
| D-glucose | Nacalai Tesque | Cat#16806-25 |
| $NaHCO_3$ | Nacalai Tesque | Cat#31213-15 |
| $NaH_2PO_4$ | Merck KGaA | Cat#106346 |
| $CaCl_2$ | Sigma-Aldrich | Cat#C5080 |
| $MgCl_2$ | Sigma-Aldrich | Cat#M2670 |
| *Drosophila* strains | | |
| *brp::SNAP* [27] | Bloomington *Drosophila* stock center (BDSC) | BDSC 58397 |
| *MB008B-GAL4* [10] | BDSC | BDSC 68291 |
| *MB009B-GAL4* [10] | BDSC | BDSC 68292 |
| *MB370B-GAL4* [10] | BDSC | BDSC 68319 |
| *VT64246-GAL4* [84] | VDRC | VDRC 204311 |
| *GH146-GAL4* [85] | BDSC | BDSC 30026 |
| *TH-GAL4* [86] | BDSC | BDSC 8848 |
| *R58E02-GAL4* [87] | BDSC | BDSC 41347 |
| *R13F02-GAL4* [87] | BDSC | BDSC 48571 |
| *OK371-GAL4* [88] | BDSC | BDSC 26160 |
| *UAS-Brp::mCherry::GCaMP6s* [35] | BDSC | BDSC 77131 |
| *cac::sfGFP* | Gratz and colleagues [6] | N/A |
| *Tdc2-GAL4* [89] | BDSC | BDSC 9313 |
| *MB022B-GAL4* [10] | BDSC | BDSC 68298 |
| *UAS-nSyb::CLIP* [27] | BDSC | BDSC 58385 |
| *UAS-DopR1-RNAi* [54] | BDSC | BDSC 55239 |
| *UAS-Dop2R-RNAi* [54] | BDSC | BDSC 50621 |
| *UAS-DopR2-RNAi* [54] | BDSC | BDSC 51423 |
| *UAS-OAMB-RNAi* [48] | VDRC | VDRC 2861 |
| *UAS-Octα2R-RNAi* [54] | BDSC | BDSC 50678 |
| *UAS-Octβ1R-RNAi* [54] | BDSC | BDSC 58179 |
| *UAS-Octβ2R-RNAi* [54] | BDSC | BDSC 34673 |
| *UAS-Octβ3R-RNAi* [54] | BDSC | BDSC 62283 |
| *UAS-TyrR-RNAi* [54] | BDSC | BDSC 25857 |
| *UAS-TyrR2-RNAi* [54] | BDSC | BDSC 64964 |

*(Continued)*

**Table 1.** (Continued)

| Reagent or resource | Source | Identifier |
|---|---|---|
| UAS-OctTyrR-RNAi [54] | BDSC | BDSC 28332 |
| UAS-Rut-RNAi [54] | BDSC | BDSC 80468 |
| UAS-CD4::tdTomato | BDSC | BDSC 35841 |
| UAS-CD8::GFP | BDSC | BDSC 32194 |
| Ay-Gal4 [33] | BDSC | BDSC 4413 |
| hs-flp$^{122}$ [90] | From Dr. Hiroyuki Uechi | |
| $T\beta h^{SK1-8}$ | This study | N/A |
| $T\beta h^{SK2-4}$ | This study | N/A |
| brp::GFP$_{11}$ | This study | N/A |
| UAS-GFP$_{1-10}$ | Kondo and colleagues [26] | N/A |
| UAS-myr::GCaMP6s | Akbergenova and colleagues [5] | |
| Software | | |
| Python | N/A | N/A |
| GraphPad PRISM | GraphPad software | https://www.graphpad.com/ |
| Fiji/ImageJ | NIH | https://imagej.net/software/fiji/downloads |
| Amira Software | Thermo Fisher Scientific | thermofisher.com/amira-avizo |
| DeconvolutionLab2 | Sage and colleagues [91] | https://bigwww.epfl.ch/deconvolution/deconvolutionlab2/ |
| CLIJ2 | Haase and colleagues [92] | https://clij.github.io/ |
| 3D ImageJ Suite | Ollion and colleagues [93] | https://imagej.net/plugins/3d-imagej-suite/ |

G9023)-0.1% PBT blocking solution for 1 hour. The nc82 antibody (see Table 1) solution was diluted 1:20 in the blocking solution. Samples were incubated in antibody solutions at 4 °C for 48 h for both primary and secondary antibodies. Sub-resolution fluorescent beads (Tetraspeck Microspheres 0.1 mm, Thermo Fisher Scientific, T7279) were imaged for generating experimental point spread function (PSF). The beads solution was diluted with distilled water and sonicated multiple times to eliminate aggregation.

For the in vivo live imaging of PM-Mns in developing pupae, samples were prepared as previously described [31]. In brief, white prepupal stage pupae (~ 0 h after puparium formation (APF)) were collected and incubated at 25 °C until the experiment. Before imaging, the puparium case was carefully removed with forceps, and the pupa was transferred to an imaging chamber sealed with a cover slip to create an imaging window on the abdomen. For the ex vivo live imaging, the 3rd instar pupal CNS was dissected directly from the animal in PBS. The CNS sample was then mounted with a spacer between the microscopic glass and cover slip in PBS. The entire process, from dissection to imaging, was performed within 5 min.

## Image acquisition

For fixed sample and the ex vivo imaging, images are acquired using the Olympus FV1200 confocal microscope platform equipped with GaAsP high sensitivity detectors and a 60×/1.42 NA oil immersion objective (PLAPON60XO, Olympus) and a 30×/1.05 NA silicone immersion objective (UPLSAPO30XS, Olympus). For the in vivo imaging on pupae, images were acquired using the Zeiss LSM 800 series confocal microscopy equipped with a 40×/1.30 NA objective (Plan-Apochromat 40×/1.30 Oil DIC (UV) VIS-IR M27). For STED microscopy, images were acquired on the Leica DMI8-CS inverted microscope STELLARIS confocal platform. A HC PL APO CS2 100×/1.40 NA oil immersion objective (Leica) was used, with a voxel size of 0.018 × 0.018 × 0.182 μm. A 489 nm laser was used for excitation and a 589 nm laser was used for depletion.

For Brp::SNAP chemical tagging, 30×/1.05 NA objective is used with a voxel size of 0.53 μm × 0.53 μm × 0.84 μm (lateral × lateral × axial) to image the whole brain. For Brp::rGFP and *UAS-CD4::tdTomato* samples in all GAL4 types except *Tdc2-GAL4*, 60×/1.42 NA oil immersion objective was used and 473 nm laser power: 0.1%; 559 nm laser power: 0.1%; scanning speed: 2.0 μs/pixel with a voxel size of 0.079 × 0.079 × 0.370 μm. For Brp::SNAP/Cac::sfGFP images, 60×/1.42 NA objective is used with a voxel size of 0.079 × 0.079 × 0.370 μm. For *Tdc2-GAL4* driven *UAS4-CD8::GFP* and *UAS-nSyb::CLIP* samples, 30×/1.05 NA objective is used with a voxel size of 0.414 × 0.414 × 0.84 μm. For MB022B-GAL driven *UAS-CD8::GFP* and *UAS-nSyb-CLIP* samples, 30×/1.05 NA objective is used with a voxel size of 0.276 × 0.276 × 0.87 μm. For *Tdc2-GAL4* driven Brp::rGFP and *UAS-CD4::tdTomato*, 60×/1.42 NA objective is used with a scanning voxel size of 0.132 × 0.132 × 0.34 μm. For Brp::rGFP *in vivo* imaging of PM-Mns, a 40×/1.3 NA objective is used with a voxel size of 0.0725 × 0.0725 × 0.460 μm. For Brp::rGFP *ex vivo* imaging, 60×/1.42 NA objective is used with a voxel size of 0.079 × 0.079 × 0.370 μm and 473 nm laser power: 2.0%. For experimental PSF imaging, SeeDB2S immersed beads were scanned in a setting that is 60×/1.42 NA oil immersion objective; 473 nm laser power: 2.0%; 559 nm laser power: 1.5%; voxel size: 0.079 μm × 0.079 μm × 0.370 μm; scanning speed: 4 μs/pixel, to produce multiple bead images.

### *In vivo* calcium imaging

*In vivo* calcium imaging was performed following a previously described protocol [96]. Briefly, flies were anesthetized on ice for 3 min and placed in a custom-made holding dish on a Peltier plate (CP-085, Sinics) maintained at 5 °C. The head capsule was fixed to the dish using UV curing optical adhesive (NOA68, Thorlabs). To minimize brain movement, the pro-boscis was glued to the capsule.c A small window was opened on the top of the head capsule, and the exposed area was filled with *Drosophila* saline solution (final concentration: 103 mM NaCl, 3 mM KCl, 5 mM TES, 8 mM Trehalose dihydrate, 10 mM D-glucose, 26 mM NaHCO$_3$, 1 mM NaH$_2$PO$_4$, 1.5 mM CaCl$_2$, 4 mM MgCl$_2$, adjust to pH ~ 7.2). Air sacs and fat bodies covering the brain surface were carefully removed. Live imaging was conducted using a laser scanning confocal micro-scope equipped with GaAsP detectors (A1R, Nikon) and a 25×/1.10 NA water immersion objective (Apo LWD 25×, Nikon). GCaMP6s and mCherry/tdTomato were excited at 488 and 561 nm, respectively. Emission was collected using dichroic mirrors and emission filters (BP500–550 and BP570–620) onto GaAsP detectors. The frame rate was set to 1 Hz.

### Data processing and analysis

PSF images were processed in the Amira software (Thermo Fisher Scientific). Using the Extract Point Spread Function module, PSFs extracted from each image were averaged into a single PSF, which was later used for image deconvolution (resized voxel size: 0.079 μm × 0.079 μm × 0.370 μm; image size: 32 pixels × 32 pixels × 21 slices). Image deconvolution was performed using the Richardson-Lucy iterative non-blind algorithm in the Fiji plugin DeconvolutionLab2 [91], or with CLIJ2 GPU-based Richardson–Lucy deconvolution [92].

AZ detection was performed as described [30] in a compartmental manner. In brief, a sub-image stacks were cropped from every MB compartment in the original image. Compartments were manually identified referencing the CD4::tdTomato membrane signal. An intensity threshold was applied to reduce background noise. Sub-stacks were processed by the 3D maxima finder and the 3D spot segmentation function in the 3D suite plugin [93] in Fiji. 3D maxima were detected for each Brp::rGFP cluster and used as start points for pixel clustering using the 3D spot segmentation function. This process created 3D ROIs enclosing individual Brp clusters. ROIs were then used to extract Brp::rGFP signal intensities independently for each cluster. Detection precision was optimized by comparing detection results with manually defined ground truths as previously described [30].

For calcium imaging, ROIs were manually drawn for each compartment in time-series image stacks. GCaMP6s signal was normalized by mCherry/CD4::tdTomato in each timeframe using the image calculator in Fiji. The mean intensity was calculated in each compartment by averaging multiple frames without any odor stimulation.

## Statistical analysis

Statistical analyses were performed using GraphPad Prism version 8, 9 and 10. Summarized data are represented as box plots showing center (median) and whiskers (Min. to Max.) or bars (mean) and whiskers (S.E.M) for nonparametric or parametric tests, respectively. Statistical tests were indicated in corresponding legends. The original False Discovery Rate (FDR) method of Benjamini and Hochberg correction was used for post-hoc multiple comparisons. Desired false discovery rate was set to 0.05. Pearson's correlation coefficient (R) was calculated using Python or Prism.

## Supporting information

**S1 Data. Data underlying Figs 2–6, S1, S6, S8–S12.**
(XLSX)

**S1 Fig. Heterogeneous Brp enrichment in the MB. (A)** SNAP chemical tagging labeled endogenous Brp in the brain. Scale bar, 100 μm. The dashed line area indicates the zoomed-in area shown in (B). **(B)** Intensity difference of Brp::SNAP between γ1 and γ5 compartment on the same imaging plane. Scale bar, 50 μm. **(C)** Schematic of the MB. The MB comprises three lobes based on the projection patterns of three KC subtypes: γ KCs, α′/β′ KCs and α/β KCs. **(D)** Brp::SNAP intensity difference across compartments. Schematic drawings above indicate compartments of each lobe. Error bars show S.E.M. Significant differences ($P < 0.05$) are indicated by distinct letters. Repeated measures one-way ANOVA. The data underlying this Figure can be found in S1 Data.
(TIFF)

**S2 Fig. Pre-synapses in the MB are predominantly from KCs.** Pie charts showing the composition of pre-synapses in each of the three MB lobes. Data is from the hemibrain online data (not all the cell types are listed). The number of pre-synapse and the percentage are indicated for each cell type, including KCs, MBONs PPL-1 neurons, PAM neurons, dorsal paired medial (DPM) neuron and anterior paired lateral (APL) neuron.
(TIFF)

**S3 Fig. Confocal and STED images of Brp::rGFP in 3rd instar larval motor neuron terminals.** White arrows indicate single AZs that show donut-shape in the STED image. Scale bar, 1 μm in the overviews; 500 nm in the insets.
(TIFF)

**S4 Fig. Brp compartmental heterogeneity in different KC subtypes.** The 3D reconstruction of Brp::rGFP clusters, colored by Brp::rGFP intensity. Brp::rGFP is visualized in γ KCs using *MB009B-GAL4*, in α/β KCs using *MB008B-GAL4* and in α′/β′ KCs using *MB0370B-GAL4*. Scale bars, 20 μm.
(TIFF)

**S5 Fig. CD4::tdTomato and Brp::rGFP in γ5 and γ1 imaged on the same focal plane.** Two independent samples are shown. While tdTomato intensities are similar in both compartments, Brp::rGFP is weaker in γ1, indicating the Brp heterogeneity is not due to imaging depth difference. Upper panels, single slice in the image stack where both γ5 and γ1 are shown. White boxes indicate zoomed-in areas shown in the lower panels. Scale bar, upper panels: 10 μm; lower panels: 2 μm.
(TIFF)

**S6 Fig. Brp is associated with Ca²+ channels in the fly brain. (A)** Co-labeling of Cac::sfGFP (green) and Brp::SNAP (magenta) in an adult brain. The write box indicates the zoomed-in areas shown in (B). Scale bar, 50 μm. **(B)** Co-localization of Brp::SNAP and Cac::sfGFP signals. Signal intensity profiles of both Brp::SNAP (green) and Cac::sfGFP (magenta) in the image were plotted below. Scale bar, 10 μm. **(C)** Correlation between Brp::SNAP (green) and Cac::sfGFP (magenta) signal intensities. The image is a selected area from γ5. Yellow circles show the 3D ROIs generated by

segmenting Brp::SNAP signals. A loose setting was applied to include surrounding pixels. The same ROI set was used to quantify the signal density (total grey value divided by the ROI volume) for both Brp::SNAP and Cac::sfGFP. **(D)** Scatter plot showing the correlation between Brp::SNAP and Cac::sfGFP signal intensities in an image sample. A 180° rotated Cac::sfGFP image was used as a control (see also S7 Fig). Pearson's correlation coefficient (*R*) is shown. **(E)** Pearson's correlation coefficient (*R*) from three individual γ lobes showing the correlation between Brp::SNAP and Cac::sfGFP signal intensities. Data are represented as box plots showing center (median), whiskers (Min. to Max.). Significant differences ($P < 0.05$) are indicated by distinct letters. Kruskal–Wallis test. The data underlying this Figure can be found in S1 Data. (TIFF)

**S7 Fig. Correlation between Brp::SNAP and Cac::sfGFP signals intensities. (A)** Correlation analysis of Brp::SNAP and Cac::sfGFP signal intensities. ROIs are generated using 3D spot segmentation method without a watershed process. Relatively loose setting was applied on Brp::SNAP images to generate wide ROIs. The same ROI set was used to calculate signal intensities in both Brp::SNAP and Cac::sfGFP channels. The 180° rotated version of Cac::sfGFP image was used as the control. A total of 4,000 ROIs were analyzed, and Pearson's correlation coefficient (*R*) values were calculated for each sample. Scale bars, 5 μm. **(B)** Scatter plots showing the correlation between Brp::SNAP and Cac::sfGFP signal intensities (left) and control (right) in different brain samples. R value is indicated for each sample. (TIFF)

**S8 Fig. Basal Ca²⁺ concentrations in different γ compartments measured by myr::GCaMP6s.** The ratio of myr::G-CaMP6s to CD4::tdTomato in γ compartments is shown. Experimental procedures and quantification are comparable to that of Fig 2E. Error bars show S.E.M. Significant differences ($P < 0.05$) are indicated by distinct letters. Repeated measures one-way ANOVA. The data underlying this Figure can be found in S1 Data. (TIFF)

**S9 Fig. AZ number/density difference along γ lobe compartments. (A)** Pre-synapse (AZ) number in each γ compartment annotated in the hemibrain connectome for γ-main KCs, γ-dorsal KCs and combined. All γ KCs annotated in the data set are quantified. γ-m KCs, $n = 588$; γ-d KCs, $n = 99$. Box plots showing center (median), whiskers (Min. to Max.). **(B)** AZ density (#Brp cluster/CD4::tdTomato area) quantified in different compartments using our image analysis pipeline. The same data set as in Fig 3B is used to quantify. $n = 11$ brains. Error bars show S.E.M. The data underlying this Figure can be found in S1 Data. (TIFF)

**S10 Fig. Dopamine receptors knockdown does not affect the Brp compartmental heterogeneity. (A)** Schematic showing the innervation patterns of dopamine neurons. The γ lobe is innervated by dopamine neurons from the PPL1 and PAM clusters. **(B)** Knockdown of dopamine receptors does not significantly alter the Brp heterogeneity level. Three types of DA receptors, DopR1, DopR2 and D2R were knocked down using RNAi in KCs specifically with *R13F02-GAL4*. Representative images of Brp::rGFP from each group are shown. Pseudo color in the images represents the value of $\log_2$ (pixel intensity/mean pixel intensity in the γ lobe). Pseudo color range: −1.3 to 1.3. Scale bar: 20 μm. Ctrl ($n = 24$) vs. DopR1 ($n = 7$), DopR2 ($n = 7$) and D2R ($n = 10$): $P > 0.05$. Box plots showing center (median), whiskers (Min. to Max.). ns = not significant by Kruskal–Wallis test. The data underlying this Figure can be found in S1 Data. (TIFF)

**S11 Fig. Maturation of compartmental heterogeneity of Brp clusters in adult KCs requires *Tβh*.** Brp::rGFP heterogeneity levels were measured at different time points after eclosion for Ctrl and *Tβh* mutants. *Tβh* mutants show impaired maturation of the compartmental Brp heterogeneity in early adulthood. **(A)** Brp::rGFP heterogeneity level measured at 3 hours and 5 days after eclosion. Ctrl 3 h ($n = 10$) vs. *Tβh^{SK1-8}* 3 h ($n = 10$): $P > 0.05$; Ctrl 3h vs. *Tβh^{SK2-4}* 3 h ($n = 11$): $P > 0.05$;

Ctrl 5 d ($n = 15$) versus $T\beta h^{SK1-8}$ 5 d ($n = 12$): $P = 0.0333$; Ctrl 5 d vs. $T\beta h^{SK2-4}$ 5 d ($n = 15$): $P = 0.0333$. Error bars show S.E.M. Kruskal–Wallis test. $*P < 0.05$ and ns = not significant. Pseudo color in the images represents the value of $\log_2$ (pixel intensity/mean pixel intensity in the γ lobe). Pseudo color range: −1.2 to 1.2. **(B)** Median Brp::rGFP intensities in γ KCs measured at 3 h and 5 days after eclosion. Box plots showing center (median), whiskers (Min. to Max.). The data underlying this Figure can be found in S1 Data.
(TIFF)

**S12 Fig. Brp::rGFP intensity and the calculated Brp heterogeneity level in each experiment. (A)** Fed, starvation and refeeding in Fig 3D. **(B)** Dopamine receptors knockdown in S10 Fig. **(C)** Ctrl and $T\beta h$ comparison in Fig 4E. **(D)** Octopamine receptors knockdown in Fig 5A. **(E)** Tyramine receptors knockdown in Fig 5A. **(F)** Rut knockdown in Fig 5B. **(G)** Food starvation in Fig 6B. **(H)** Starvation of $T\beta h^{SK2-4}$ mutants in Fig 6B. **(I)** Food sleep deprivation in Fig 6C. **(J)** Sleep deprivation of $T\beta h^{SK2-4}$ mutants in Fig 6C. Box plots showing center (median), whiskers (Min. to Max.). See sample number and statistics of heterogeneity level comparison in legends of main figures. The data underlying this Figure can be found in S1 Data.
(TIFF)

## Acknowledgments

We thank all lab members of Tanimoto Lab at Tohoku University and Williams Lab at King's College London for valuable discussions. We thank Dr. Kate O'Connor-Giles (Brown University), Dr. Troy Littleton (Massachusetts Institute of Technology), Vienna *Drosophila* Resource Center (VDRC) and Bloomington *Drosophila* Stock Center (BDRC) for transgenic flies, Nobuhiro Takahashi for the sleep deprivation system, Ayano Wu (Tohoku University) for graphical design.

## Author contributions

**Conceptualization:** Hongyang Wu, Nobuhiro Yamagata, Hiromu Tanimoto.

**Formal analysis:** Hongyang Wu, Sayaka Eno.

**Funding acquisition:** Hongyang Wu, Darren W. Williams, Hiromu Tanimoto.

**Investigation:** Hongyang Wu, Sayaka Eno, Kyoko Jinnai, Ayako Abe, Kokoro Saito.

**Methodology:** Hongyang Wu, Sayaka Eno, Yoh Maekawa, Darren W. Williams, Nobuhiro Yamagata, Shu Kondo.

**Project administration:** Hiromu Tanimoto.

**Resources:** Kyoko Jinnai, Shu Kondo, Hiromu Tanimoto.

**Supervision:** Darren W. Williams, Nobuhiro Yamagata, Hiromu Tanimoto.

**Validation:** Hongyang Wu, Sayaka Eno.

**Visualization:** Hongyang Wu, Sayaka Eno.

**Writing – original draft:** Hongyang Wu, Sayaka Eno, Hiromu Tanimoto.

**Writing – review & editing:** Hongyang Wu, Sayaka Eno, Kyoko Jinnai, Ayako Abe, Kokoro Saito, Yoh Maekawa, Darren W. Williams, Nobuhiro Yamagata, Shu Kondo, Hiromu Tanimoto.

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
