## [Editor Report · Decision Letter 0]

21 Aug 2025

Dear Hiromu,

Thank you for submitting your manuscript entitled "Octopamine signals regulate the intracellular pattern of the presynaptic active zone scaffold within Drosophila mushroom body neurons" for consideration as a Research Article by PLOS Biology. Please allow me first to apologize for the long delay in getting back to you. Due to the holiday season it took much longer than we would have liked to discuss your revised manuscript within the team and the Academic Editor.

Your manuscript has now been evaluated by the PLOS Biology editorial staff as well as by an academic editor with relevant expertise and I am writing to let you know that we would like to send your manuscript back to external peer review.

Once your full submission is complete, your paper will undergo a series of checks in preparation for peer review. After your manuscript has passed the checks it will be sent out for review. To provide the metadata for your submission, please Login to Editorial Manager (https://www.editorialmanager.com/pbiology) within two working days, i.e. by Aug 23 2025 11:59PM.

Kind regards,

Christian

Christian Schnell, Ph.D.

Senior Editor

PLOS Biology

cschnell@plos.org

on behalf of

Taylor Hart, PhD,

Associate Editor

PLOS Biology

thart@plos.org

---

## [Decision Letter · Decision Letter 1]

18 Sep 2025

Dear Dr Tanimoto,

Thank you for your patience while we considered your revised manuscript "Octopamine signals regulate the intracellular pattern of the presynaptic active zone scaffold within Drosophila mushroom body neurons" for publication as a Research Article at PLOS Biology. This revised version of your manuscript has been evaluated by the PLOS Biology editors, the Academic Editor, and two of the original reviewers.

Based on the reviews and on our Academic Editor's assessment of your revision, we are likely to accept this manuscript for publication. Please also make sure to address the following data and other policy-related requests.

IMPORTANT: Please ensure that your next revision addresses the following points:

-------------

**Title

We suggest a slight tweak of your title to the following: "Octopamine signaling regulates the intracellular pattern of the presynaptic active zone scaffold within Drosophila mushroom body neurons"

**Data:

-- We see that you wrote that the data files will be made available from the German Infrastructure Node. Please make them available and provide a link so that we can examine these files prior to publication.

-- As part of this, please supply the numerical values either in a supplementary excel file or as a permanent DOI’d deposition for the following figures panels:

2DE

3BCD

4CE

5AB

6BC

S1D

S6E

S8

S9AB

S10B

S11AB

S12ABCDEFGH

-- Please cite the location of the data clearly in all relevant main and supplementary Figure legends, e.g. “The data underlying this Figure can be found in S1 Data” or “The data underlying this Figure can be found in https://doi.org/10.5281/zenodo.XXXXX”

-- Supplementary files (e.g., excel). Please ensure that all data files are uploaded as 'Supporting Information' and are invariably referred to (in the manuscript, figure legends, and the Description field when uploading your files) using the following format verbatim: S1 Data, S2 Data, etc. Multiple panels of a single or even several figures can be included as multiple sheets in one excel file that is saved using exactly the following convention: S1_Data.xlsx (using an underscore).

-------------

We expect to receive your revised manuscript within two weeks.

*Published Peer Review History*

*Press*

Sincerely,

Taylor

Taylor Hart, PhD,

Associate Editor

thart@plos.org

PLOS Biology

Reviewer remarks:

Reviewer #1: The authors have included substantial text revisions and new experimental data in this submission in response to critiques from each reviewer. In particular, the addition of BRP distribution data from individual gamma KCs significantly strengthens the manuscript and supports several of the authors' conclusions. The authors have substantially addressed the previous comments and I have no further issues to raise.

Reviewer #2: The authors have addressed the issues raised in my prior critique and I have no further concerns.

---

## [Editor Report · Decision Letter 2]

3 Oct 2025

Dear Dr Tanimoto,

Thank you for the submission of your revised Research Article "Octopamine signaling regulates the intracellular pattern of the presynaptic active zone scaffold within Drosophila mushroom body neurons" for publication in PLOS Biology. On behalf of my colleagues and the Academic Editor, Bing Ye, I am pleased to say that we can in principle accept your manuscript for publication, provided you address any remaining formatting and reporting issues. These will be detailed in an email you should receive within 2-3 business days from our colleagues in the journal operations team; no action is required from you until then. Please note that we will not be able to formally accept your manuscript and schedule it for publication until you have completed any requested changes.

Please note that I have made a minor change to your Supplementary Data -- adding the label "Fig. 6C" where it appeared to be missing in the original version.

PRESS

Sincerely, 

Taylor Hart, PhD,

Associate Editor

PLOS Biology

thart@plos.org